# *Lacticaseibacillus casei* IB1 Alleviates DSS-Induced Inflammatory Bowel Disease by Regulating the Microbiota and Restoring the Intestinal Epithelial Barrier

**DOI:** 10.3390/microorganisms12071379

**Published:** 2024-07-06

**Authors:** Jianlong Lao, Shuping Yan, Yanhong Yong, Yin Li, Zhaohai Wen, Xiaoyong Zhang, Xianghong Ju, Youquan Li

**Affiliations:** 1College of Coastal Agricultural Sciences, Guangdong Ocean University, Zhanjiang 524088, China; jianlong_lao@163.com (J.L.); yanshuping@gdou.edu.cn (S.Y.); yongyanhong-007@163.com (Y.Y.); ly12181028@163.com (Y.L.); wenzhaohai@gdou.edu.cn (Z.W.); zhangyong970520@163.com (X.Z.); 2Marine Medical Research and Development Centre, Shenzhen Institute of Guangdong Ocean University, Shenzhen 518120, China

**Keywords:** probiotics, inflammatory, signaling pathway, tight junction, gut microbiota

## Abstract

Inflammatory bowel disease (IBD) is becoming an increasingly serious health problem in humans and animals. Probiotics can inhibit the development of IBD. Due to the specificity of the strains, the function and mechanism of action of different strains are still unclear. Here, a DSS-induced colitis mouse model was utilized to investigate the ability and mechanism by which *Lacticaseibacillus casei* IB1 alleviates colitis. Treatment with *L. casei* IB1 improved DSS-induced colitis in mice, as indicated by increased body weight, colon length, and goblet cell numbers and decreased disease activity index (DAI), proinflammatory factor (TNF-α, IL-1β, and IL-6) levels, and histopathological scores after intake of IB1. IB1 supplementation also improved the expression of tight junction proteins and inhibited the activation of the MAPK and NF-κB signaling pathways to alleviate intestinal inflammation. In addition, IB1 rebalanced the intestinal microbial composition of colitis mice by increasing the abundance of *Faecalibaculum* and *Alistipes* and decreasing the abundance of *Bacteroides* and *Escherichia_Shigella*. In summary, *L. casei* IB1 showed great potential for relieving colitis by regulating the microbiota and restoring the epithelial barrier. It can be used as a potential probiotic for the prevention and treatment of UC in the future.

## 1. Introduction

Inflammatory bowel disease (IBD), comprising ulcerative colitis (UC) and Crohn’s disease (CD) [1,2,3], is an idiopathic, chronic, and frequently recurrent condition impacting the gastrointestinal tract, with global implications for human health [4]. The main symptoms of UC are abdominal pain, diarrhea, weight loss, and bloody stool [5,6,7]. There are many pathogenic factors of UC. It is generally believed that heredity, diet, the immune system, the intestinal barrier, and microflora may be related to the development of UC [7,8,9,10]. In recent years, some studies have shown that intestinal flora imbalance [11,12,13,14,15] and intestinal epithelial barrier damage [14,15,16] are related to the occurrence of IBD.

The gut microbiota is strongly linked to the physiological function and immune ability of the host [14]. It has a significant impact on the host during homeostasis and disease [17,18]. An imbalance in the intestinal microflora leads to an increase in pathogens and abnormalities in the intestinal flora [17,19]. The integrity of the intestinal barrier is indispensable for human and animal health [20]. Tight junction proteins play an important role in maintaining the intestinal mucosal barrier. Studies have shown that in a model of inflammatory bowel disease, the level of tight junction proteins is increased after treatment with probiotics [21,22]. In addition, studies have shown that the pathogenic bacteria of the intestinal microflora are significantly increased and that the level of tight junction proteins is significantly decreased in IBD patients and animal models [23,24,25,26,27]. At present, the clinical drugs commonly used to treat IBD mainly contain antibiotics, aminosalicylic acid, steroids, and immune inhibitors [28,29]. However, these drugs have significant side effects on humans and animals and their application is greatly limited. Therefore, safe, effective drugs with no side effects are urgently needed to treat IBD.

Qinghai is one of the places in northwestern China where ethnic minorities gather. Dairy products are part of traditional food culture and thus are indispensable. Compared with modern techniques, the traditional approach to dairy production is more ecological and natural, and the types of probiotics produced are more abundant. IB1 is derived from herdsmen’s yogurt in Qinghai, China, with a safe source and long history. In a preliminary study, we chose *L. casei* IB1 for research, which has been proven to be acid-resistant, has a high survival rate in simulated gastrointestinal experiments, and has proved that IB1 has no toxic effect on mice in vivo. Probiotics are a class of living microorganisms that can have a healthy effect on the host when the intake is sufficient [30]; they are considered to promote the health of the host by producing related beneficial enzymes, organic acids, vitamins, and bacteriocins [31]. At present, commonly used probiotics are *Lacticaseibacillus*, *Bifidobacterium*, *Enterococcus*, *Bacillus*, and *Saccharomycetes* [32]. Previous studies have shown that probiotics have antibacterial functions, regulating immune activity and antioxidation, improving the intestinal mucosal barrier, and regulating intestinal flora homeostasis [13,31,33,34]. However, probiotics have strain specificity, and the mechanism of action of different strains is still unclear [34]. The histological characteristics, clinical manifestations, location, and cytokine proliferation of the DSS-induced colitis model are very similar to those of human ulcerative colitis. The conditions and operation methods of the model are simple, low-cost, repeatable, and easy to master and promote [35,36]. Therefore, this study mainly used a DSS-induced colitis mouse model to explore the role and mechanism of *L. casei* IB1.

## 2. Materials and Methods

### 2.1. Preparation of L. casei IB1

*L. casei* IB1 was preserved in our laboratory and subsequently cultured in MRS broth medium at 37 °C for 24 h and continuously oscillated at 120 rpm. After incubation, the concentration of IB1 reached 2.1 × 10^10^ CFU/mL. After centrifugation at 5000 rpm for 5 min, the cell supernatants were discarded and the cell pellets were collected. The cells were washed twice with PBS, diluted to 1 × 10^10^ CFU/mL, and stored at −80 °C for subsequent experiments.

### 2.2. Animal Experiment Design

Thirty male C57BL/6J mice (6 weeks old, 16–18 g) were bought from Youda Biotechnology Co., Ltd. (Guangzhou, China). Dextran sodium sulfate (DSS, molecular weight: 36–50 kDa) was purchased from Dalian Meilun Biotechnology (Dalian, China). Throughout the animal experiment, all mice were kept in the Center of Animal Lab of Guangdong Ocean University at a temperature of 25 ± 2 °C and relative humidity of 50 ± 5%, with a 12 h light/dark cycle. The experimental process and technical route are shown in Figure 1. After 7 days of adaptation, the mice were randomly divided into 5 groups (*n* = 6): CON group, DSS model group, LOW group (DSS + L, 10^7^ CFU/mL), MID group (DSS + M, 10^8^ CFU/mL), and HIGH (DSS + H, 10^9^ CFU/mL) group. The specific test scheme was as follows: a. Throughout the experimental period, the control group was provided standard food and drinking water, and 200 μL of PBS was given daily. b. The DSS model group was given standard food and drinking water during the experiment, 3% (*w/v*) DSS solution was used instead of drinking water in the last week of the experiment, and 200 μL of PBS was given daily during the experiment. c. The mice in the DSS + L, DSS + M, and DSS + H groups were given standard food and drinking water during the experiment, 3% (*w/v*) DSS solution was used instead of drinking water in the last week of the experiment, and 200 μL of the corresponding concentration of strain solution was given daily during the experiment. During the experiment, the weights of the mice were recorded, and fecal occult blood was observed. The DAI of each mouse was calculated daily in a previous study [37]. The DAI score was assessed according to the parameters (Table 1), with a maximum total score of 12 points.

### 2.3. Sample Collection

On the 15th day of the experiment, the eyeball blood of all mice was collected in a tube, and the serum was collected at 3000 rpm and 4 °C after 15 min and stored at −80 °C for subsequent experiments. Then, the anesthetized mice were killed by cervical dislocation. Colon contents were collected aseptically for microbiotomic analysis. In addition, the length of the colon was measured, and a portion of the colon (0.5 cm section) was quickly removed and soaked in 4% paraformaldehyde for histological evaluation. The remaining colon was rapidly frozen with liquid nitrogen and stored at −80 °C for further analysis.

### 2.4. Serum Enzyme-Linked Immunosorbent Assay (ELISA)

Serum levels of tumor necrosis factor alpha (TNF-α), interleukin-10 (IL-10), interleukin-6 (IL-6), and interleukin-1β (IL-1β) were measured using commercially available ELISA kits (Jiangsu Meimian Industrial Co., Ltd., Yancheng, China).

### 2.5. Histopathological Examination of the Colon

Colon tissue samples were sent to Wuhan Servicebio Technology Co., Ltd. (Wuhan, China), for hematoxylin and eosin (HE) staining and periodic acid–Schiff (PAS) staining. Histological changes in the colon were observed through a conventional light microscope (Olympus, Tokyo, Japan), and images were taken digitally (Sony, Tokyo, Japan) for subsequent analysis according to Table 2.

### 2.6. Immunoblotting Analysis

Colon tissues were removed at −80 °C; total protein was extracted with RIPA lysis buffer (Strong), 1% protease inhibitor cocktail, and 1% phosphatase inhibitor cocktail (Jiangsu CWBIO Biotechnology Co., Ltd., Jiangsu, China); and the protein concentration was determined with the Synergy HTX Multi-Mode Reader (Bio-Tek, Winooski, VT, USA) according to the instructions of a BCA kit (Yeasen Biotechnology Co., Ltd., Shanghai, China). The protein samples were mixed with loading buffer, heated in a boiling water bath for 5 min, separated by SDS–PAGE (8%), and transferred to 0.22 μm polyvinylidene difluoride (PVDF) membranes (Merck Millipore, Burlington, MA, USA). The membrane was blocked with 5% (*w*/*v*) skim milk in Tris-buffered saline containing 1% Tween 20 (TBST) at room temperature for 2 h and then washed three times with TBST for 10 min each time. After that, the PVDF membranes were incubated with the primary antibodies overnight at 4 °C: NF-κBP65 (8242S, CST, 1:1000), NF-κBP-p65 (3033S, CST, 1:1000), P38 (8690S, CST, 1:1000), P-p38 (4511S, CST, 1:1000), ERK1/2 (4695S, CST, 1:1000), P-ERK1/2 (4370S, CST, 1:1000), JNK (67096S, CST, 1:1000), P-JNK (4668S, CST, 1:1000), Occludin (27260, Proteintech, Rosemont, IL, USA, 1:1000), Claudin1 (37–4900, Invitrogen, Waltham, MA, USA), Claudin2 (AF0128, Affbiotech, Cincinnati, OH, USA, 1:1000), and β-Actin (HC201, TransGen Biotech, Beijing, China, 1:1000). After washing three times for 10 min each with TBST buffer, the membrane was incubated with a horseradish peroxidase conjugate secondary antibody (TransGen Biotech, 1:5000) at room temperature for 2 h and then washed with TBST three times for 10 min each. Immunoblotting was performed on a Tanon 5200 chemiluminescence instrument (Tanon, Shanghai, China) using a Tanon High-sig ECL Western blotting substrate kit. ImageJ 1.53 software (National Institutes of Health, USA) was used to measure the density of each blot band.

### 2.7. Immunofluorescence Analysis

Colon tissue samples were sent to Wuhan Servicebio Technology Co., Ltd., for histological immunofluorescent analysis. The protein expression levels of ZO1 and Occludin in colon tissues were detected by immunofluorescence. The images were captured by an upright fluorescence microscope (Nikon, Tokyo, Japan) and processed by ImageJ software (National Institutes of Health, USA).

### 2.8. 16S rRNA Sequencing of the Colon Microbiota

The colon contents were transported to Beijing Biomarker Technology Co., Ltd. (Beijing, China), on dry ice, with 5 biological replicates per group. Microbial sequencing (16S length) was completed by a company using the PacBio sequencing platform, and all microbiome data analysis was completed on the Biomarker Microbial Diversity analysis cloud platform (www.biocloud.net), accessed on 18 January 2024.

### 2.9. Statistical Analysis

All the experimental results are from at least three independent experiments, and all the data are expressed as the mean ± standard error of the mean (S.E.M.). All the data were statistically significant according to one-way analysis of variance (ANOVA) and Student’s *t*-test in IBM SPSS Statistics 27. *p* < 0.05 was considered to indicate statistical significance. All figures and graphics were produced using GraphPad Prism 8.0 statistical software (GraphPad Software Inc., San Diego, CA, USA).

## 3. Results

### 3.1. L. casei IB1 Alleviated DSS-Induced Colitis in Mice

To survey the alleviating effect of *L. casei* IB1 on colitis in mice, experimental colitis models were generated in mice induced with 3% DSS in their water for 7 days (Figure 1). Compared to that of the CON group, the body weight of the DSS group was significantly lower (*p* < 0.05) (Figure 2A). However, compared with those in the DSS group, after the oral administration of *L. casei* IB1, the weight loss of the mice in the MID group and the HIGH group was alleviated to some extent (*p* < 0.05). Compared with that in the CON group, the DAI in the DSS group was significantly greater (*p* < 0.05) (Figure 2B). In contrast, after treatment with *L. casei* IB1, the DAI scores of DSS-treated mice in the MID and HIGH groups were significantly greater than those of DSS-treated mice (*p* < 0.05). The colon length of the DSS treatment group was significantly shorter than that of the CON group (*p* < 0.05) (Figure 2C,D). Compared with DSS, the oral administration of *L. casei* IB1 efficiently prevented DSS-induced colon shortening (*p* < 0.05).

### 3.2. Effect of L. casei IB1 on DSS-Induced Colon Histopathology

To observe the effect of *L. casei* IB1 on DSS-induced colon histopathology, we performed HE staining and PAS staining of colon tissue. As shown in Figure 3A–D, the colonic epithelial cells of the CON group were neatly arranged, the structure was complete, the villi were neat, the crypts were neatly arranged, there was a large number of goblet cells, and there was almost no inflammatory cell infiltration. After 7 days of DSS induction, compared with those in the CON group, the morphology of intestinal epithelial cells in the colon tissue of mice in the DSS group changed, the crypts were destroyed, the mucosal structure was unclear, the lamina propria and submucosa were infiltrated by a large number of inflammatory cells, and a large amount of connective tissue hyperplasia had occurred. In contrast, after treatment with *L. casei* IB1, compared to the DSS group, colon tissue injury was reduced in the MID and HIGH groups, the crypt structure was orderly, the mucosal structure was intact, and a large number of goblet cells were recovered (*p* < 0.01). Therefore, our results showed that *L. case*i IB1 had a certain preventive effect on DSS-induced colon injury.

### 3.3. Effect of L. casei IB1 on Serum Cytokine Levels

The levels of serum cytokines (TNF-α, IL-1β, IL-6 I, and L-10) were measured by ELISA. As shown in Figure 4A–D, compared with those in the CON group, the levels of the cytokines TNF-α, IL-1β, and IL-6 were significantly increased in the DSS group (*p* < 0.05), while the concentration of IL-10 was significantly decreased (*p* < 0.05). These results suggested that DSS treatment increased the secretion of proinflammatory cytokines and correspondingly reduced the level of anti-inflammatory cytokines. After treatment with *L. casei* IB1, the levels of the cytokines TNF-α, IL-1β, and IL-6 were significantly lower than those in the DSS group (*p* < 0.01), while the levels of the anti-inflammatory cytokine IL-10 were significantly increased (*p* < 0.01). The results showed that *L. casei* IB1 had anti-inflammatory effects on intestinal tissues.

### 3.4. L. casei IB1 Upregulated Tight Junction Proteins to Improve Colonic Barrier Dysfunction in DSS-Induced Colitis Mice

Intestinal epithelial tight junction (TJ) proteins play important roles in maintaining intestinal barrier integrity. To study the changes in TJ proteins in DSS-induced colitis, Western blotting was used to detect changes in Claudin1, Claudin2, and Occludin in colon tissue. As shown in Figure 5A–C and Appendix A, after DSS induction, the protein levels of Occludin, Claudin1, and Cluaudin2 in the DSS group were significantly lower than those in the CON group (*p* < 0.01). Compared to the DSS group, the levels of the Occludin, Claudin1, and Claudin2 proteins increased after the oral administration of *L. casei* IB1, among which, those in the MID group and HIGH group increased most significantly (*p* < 0.01).

In addition, the protein expression of ZO-1 and Occludin in colon tissue was detected by immunofluorescence to further evaluate the colonic barrier. As shown in Figure 6A–C, similar to the Western blot results, the expression of tight junction proteins in the DSS group was significantly lower than that in the CON group (*p* < 0.01). After treatment with IB1, the protein expression levels of ZO-1 and Occludin increased (*p* < 0.05). These results indicate that *L. casei* IB1 improves DSS-induced intestinal barrier damage by upregulating TJ proteins.

### 3.5. L. casei IB1 Inhibits the Inflammatory Response by Inhibiting the NF-κB- and MAPK-Mediated Signaling Pathways

To study the potential anti-inflammatory mechanism of *L. casei* IB1, we examined the effects of IB1 on the NF-κB and MAPK signaling pathways. These signaling pathways are important key regulators of the inflammatory process and play an important part in intestinal inflammation. As shown in Figure 7A–D and Appendix A, after DSS treatment, the expression of key proteins in the NF-κB and MAPK signaling pathways in the DSS group increased compared with that in the CON group (*p* < 0.01). Compared with that in the DSS group, the expression of the phosphorylated proteins P65 and P38 decreased after the oral administration of *L. casei* IB1 (*p* < 0.01). Interestingly, after the oral administration of *L. casei* IB1, only the protein levels of phosphorylated ERK and JNK in the MID group and the HIGH group decreased significantly, and there was no significant difference in the LOW group. These results suggest that *L. casei* IB1 may alleviate intestinal inflammation by inhibiting the activation of NF-κB and MAPK.

### 3.6. Effects of L. casei IB1 on Microbial Diversity in the Colon

To explore the relationship between intestinal inflammation and intestinal microbial diversity, we collected the colonic contents of the mice and performed 16S rRNA sequencing. The colon microbiota diversity is shown in Figure 8A. There were 223 common OTUs in the colon of each experimental group. As shown in Figure 8B–E, the ACE and Chao1 indices measure species richness. The Simpson and Shannon indices are used to measure species diversity. There was no significant difference in the Chao1 index between the groups. Compared with that of the control group, the ACE index of the DSS group decreased significantly (*p* < 0.05). Compared to the DSS group, the ACE index of the LOW group and HIGH group increased significantly (*p* < 0.05). There was no significant difference in the Simpson index between the CON and DSS groups. Compared to the DSS group, those of the LOW and MID groups decreased significantly (*p* < 0.05). However, compared with those of the CON group, the Shannon indices of the DSS group increased significantly (*p* < 0.05), and the diversity of the microbial community significantly improved after the oral administration of *L. casei* IB1, which was similar to the CON group. The results showed that the oral administration of *L. casei* IB1 could regulate intestinal homeostasis. To further study the similarity between microbial communities, we performed beta diversity analysis based on weighted UniFrac to perform principal coordinate analysis (PCoA) and nonmetric multidimensional scaling (NMDS). As shown in Figure 8F–G, there was a clear clustering separation between the DSS group and the CON group, indicating that the bacterial community composition differed between the two groups. However, after treatment with *L. casei* IB1, the bacterial colony composition overlapped with that of the CON group.

### 3.7. L. casei IB1 Changed the Structural Composition of Colon Microorganisms

As shown in Figure 9, the microbial flora of the five groups of colonic contents mainly consisted of five phyla, namely, *Firmicutes*, *Bacteroidetes*, *Proteobacteria*, *Verrucomicrobiota*, and Patescibacteria, and nine genera, namely, *Lachnospiraceae_NK4A136_group*, *unclassified Muribaculaceae*, *unclassified Clostridia_UCG_014*, *Bacteroides*, *uncultured Rumen_bacterium*, *Faecalibacterium*, *Escherichia_Shigella*, *Akkermansia*, and *Alistipes*. At the phylum level, the contents of *Verrucomicrobia*, *Firmicutes*, *Bacteroidetes*, *Proteobacteria*, and *Patescibacteria* in the CON group were 51.80%, 41.37%, 1.32%, 0.78%, and 4.17%, respectively. Compared with those in the CON group, the contents of *Firmicutes*, *Bacteroidetes*, and *Patescibacteria* in the DSS group decreased to 31.82%, 31.69%, and 0.06%, respectively, while the contents of *Proteobacteria* and *Verrucomicrobiota* increased to 20.69% and 12.67%, respectively. After treatment with *L. casei* IB1, the contents of *Proteobacteria* and *Verrucomicrobia* in the LOW group decreased to 3.38% and 0.62%, those in the MID group decreased to 2.42% and 1.72%, and those in the HIGH group decreased to 1.94% and 4.04%, respectively. At the genus level, the abundance of *Faecalibaculum* decreased after DSS treatment, while that of *Escherichia_Shigella* increased. Interestingly, after treatment with *L. casei* IB1, the abundance of *Faecalibacterium* increased, while the disruption of *Escherichia_Shigella* normalized.

LEfSe and linear discriminant analysis (LDA) were used to analyze and compare the intestinal flora of each group to better understand the indicator bacteria of each group. As shown in Figure 10A,B, there were 21, 24, 11, 7, and 6 significant differences between the CON group, DSS group, LOW group, MID group, and HIGH group, respectively. The relative abundances of *Muribaculaceae*, *Bacteroidota*, *Lactobacillaceae*, *Saccharimonadaceae* and *Prevotella* in the CON group were greater than those in the DSS group. However, the bacterial diversity that affected the DSS group’s microbiome was distributed in different units. The abundance of potential pathogenic bacteria, such as *Escherichia_Shigella* at the genus level and *Enterobacteriaceae* at the family level, increased in the DSS group. After treatment with *L. casei* IB1, the abundance of potential pathogenic bacteria decreased to the lowest level. In the LOW group, the relative abundances of *Clostridia* at the class level and *Erysipelatoclostridium* and *Alistipes* at the genus level were greater. In the MID group, the relative abundances of *Firmicutes* at the phylum level and *uncultured_rumen_bacterium* and *Turicibacter* at the genus level were greater. In contrast, the relative abundances of *unclassified _Clostridia _UCG _014* and *Faecalibaculum* at the genus level in the HIGH group were greater. In summary, treatment with *L. casei* IB1 alleviated DSS-induced colitis in mice by reducing the abundance of potential pathogenic bacteria and increasing the relative abundance of probiotics.

## 4. Discussion

Recently, great progress has been made in the use of probiotics for the treatment of IBD. Probiotics mainly regulate the intestinal barrier [16,38] and the balance of the intestinal flora [39,40,41] to alleviate IBD. At present, the DSS-induced colitis mouse model is a useful tool for studying the efficacy and mechanism of probiotics in relieving acute IBD [33,35,36,42]. The mechanism of probiotic treatment for IBD may be related to reducing oxidative stress, repairing the intestinal barrier, regulating the intestinal flora balance, and modulating the intestinal immune response [43]. As one of the representative bacteria for probiotic-assisted therapy in IBD, multiple strains of *Lactobacillus* have been proven to alleviate intestinal damage and strengthen the intestinal immunological barrier, epithelial cell barrier, and mucus barrier [44]. *Lactic* acid bacteria are recognized as safe for use in food fermentation and dietary supplementation and are also thought to potentially colonize the human gastrointestinal tract [45]. Therefore, we used a DSS-induced mouse colitis model to study the effect of *L. casei* IB1 on colitis in C57BL/6 mice. After 7 days of 3% DSS treatment, the mice showed severe damage, and the clinical symptoms were weight loss, diarrhea, and rectal bleeding [23,46,47]. In addition, we confirmed that DSS significantly increased the DAI. On the contrary, the symptoms of colitis in mice treated with *L. casei* IB1 were significantly alleviated. Compared with the DSS group, treatment with *L. casei* IB1 significantly increased the colon length of the mice. Histopathological evaluation of the colon tissue revealed that *L. casei* IB1 treatment significantly reduced colonic mucosal damage, crypt injury, inflammatory cell infiltration, and local ulcers, thereby restoring the integrity of the intestinal epithelium. Goblet cells are glandular cells that secrete mucus, lubricate the surface of the epithelium, and protect the epithelium. They can perform lubrication and physical barrier functions to prevent pathogenic microorganisms from invading the host [48,49]. In this study, compared with that in the DSS group, the number of goblet cells increased after treatment with *L. casei* IB1, especially in the MID and HIGH groups. These results confirmed the therapeutic potential of *L. casei* IB1.

Cytokines have a wide range of biological activities. They modulate cell differentiation and growth by binding to corresponding receptors and regulating immune responses [50]. Previous studies have shown that the levels of proinflammatory factors increase and the levels of anti-inflammatory factors decrease in patients with ulcerative colitis and DSS-induced colitis in animal models [51,52,53]. Inflammatory factors such as IL-6, IL-10, and TNF-α are the main immune response factors involved in the inflammatory response in IBD [54]. Some researchers believe that blocking IL-1β and IL-6 may be an important direction for the treatment of UC [55]. The concentration of cytokines in serum represents the systemic inflammatory response [55]. In this study, we measured the serum levels of pro-inflammatory factors (TNF-α, IL-1β, IL-6) and the anti-inflammatory factor IL-10 using ELISA. Compared to the DSS group, the levels of pro-inflammatory factors were significantly decreased, while the level of IL-10 was markedly increased after treatment with *L. casei* IB1. These results suggest that *L. casei* IB1 may mitigate colitis by reducing proinflammatory factor levels and increasing anti-inflammatory factor levels.

The integrity of the gut barrier is mainly supported by the tight junctions of epithelial cells. The integrity of the intestinal barrier can prevent pathogens from entering the blood [47,56]. Studies have shown that DSS intervention can reduce the concentration of related tight junction proteins [57]. ZO-1, occludin, and claudin-1 are important epithelial TJ proteins [56,58]. In this study, Western blotting and immunofluorescence were used to detect related TJ proteins. Compared with those in the DSS group, the levels of related TJ proteins were greater after treatment with *L. casei* IB1, and the increase in the middle and high groups was the most significant. The results suggested that *L. casei* IB1 alleviated colitis by improving the expression of TJ proteins.

The NF-κB and MAPK signaling pathways are involved in the pathogenesis of IBD [59,60]. Therefore, the inhibition of NF-κB and MAPK activation is considered to be an important target for the treatment of IBD [61]. The MAPK pathway, which includes the JNK, ERK, and p38 proteins, can regulate different physiological processes (apoptosis, proliferation, and differentiation) in cells and is closely linked to intestinal mucosal injury [48]. These results suggest that *L. casei* IB1 may alleviate DSS-induced colitis by inhibiting MAPK and NF-κB signaling pathway-related proteins.

The intestinal microflora plays a very important role in maintaining human health, and the stability of the microbial community is conducive to the health of the host [62]. Intestinal microbial imbalances have been shown to be related to various diseases [63]. The gut microbiota has been identified as a key factor in the pathogenesis of IBD [47]. Therefore, the use of probiotics to prevent or treat IBD is a potential treatment strategy. In this study, 16S rRNA sequencing was used to explore the mechanism of action of *L. casei* IB1 on the intestinal microflora in DSS-induced colitis mice. As with previous studies [55], a wide range of gut microbiota disorders were observed, including a decrease in the number of OTUs and a disorder of β-diversity. The Chao1 and ACE indices measure species richness, that is, the number of species. The Shannon and Simpson indices are used to measure species diversity. The ACE and Chao1 indices decreased under the influence of DSS, while the Simpson and Shannon indices increased. However, these effects were reversed after treatment with *L. casei* IB1. Through PCoA and NMDS cluster analysis, it could be seen that the CON group and DSS group were obviously separated and, after treatment with *L. casei* IB1, the LOW group, MID group, and HIGH group overlapped with the CON group. In summary, *L. casei* IB1 alleviates colitis by regulating the abundance and diversity of microorganisms.

In UC patients, intestinal microbial diversity changes significantly, and intestinal microbial abundance decreases. When intestinal inflammation occurs, the proportion of *Firmicute*s that maintain intestinal health decreases and the abundance of *Proteobacteria* increases [52,61,64]. Consistent with the above reports, in our study, the proportion of *Firmicutes* was reduced and the abundance of *Proteobacteria* was increased in DSS-induced colitis mice. *Escherichia_Shigella* is a common pathogen that often increases in patients with colitis [65], and *Alistipes* has been shown to have the opposite relationship with a variety of inflammatory factors [62]. Bacteroides can produce succinic acid, which aggravates the inflammatory response in patients with ulcerative colitis [62,66]. Some studies have found that *Faecalibaculum rodentium* remodels retinoic acid signaling to govern eosinophil-dependent intestinal epithelial homeostasis [67]. *Faecalibaculum* was reported to protect the intestinal epithelial barrier by producing butyric acid [62,68]. Consistent with the above report, our study found that after treatment with IB1, the abundance of *Faecalibaculum* was upregulated, which may have led to the production of more butyric acid to mitigate colitis. *Escherichia Shigella* and *Bacteroides* exhibited a higher relative abundance in IBD individuals and further led to severe colitis [66,69]. Consistent with the above reports, the abundances of *Escherichia Shigella* and *Bacteroides* in the DSS group increased at the genus level compared with those in the CON group. However, after treatment with *L. casei* IB1, the proportions of *Bacteroides* and *Escherichia Shigella* decreased and the *proportions of Faecalibacterium* and *Alistipes* increased. The intestinal environment in vivo is more complex than the cell culture environment in vitro. There are interactions among microorganisms, microbial immune cells, and microbial intestinal epithelial cells in the intestinal tract [14]. IB1 may protect the intestinal barrier by interacting with other bacteria or influencing immune cells. It has been reported that the intestinal symbiotic flora can reduce the susceptibility of mice to experimental colitis through T-cell-derived IL-10 [70]. Therefore, *L. casei* IB1 promoted the growth of beneficial bacteria in the intestine tract, especially beneficial microorganisms related to butyric acid production, and inhibited harmful microorganisms related to intestinal diseases. In this study, IB1 alleviated DSS-induced colitis in mice to a certain extent by improving the intestinal barrier, inhibiting the inflammatory response, and reshaping the intestinal microbiome structure. Further research is needed to investigate the mechanism of IB1 in the prevention and treatment of IBD.

## 5. Conclusions

In summary, *L. casei* IB1 can effectively improve DSS-induced colitis in mice. Its effects include alleviating the clinical symptoms caused by DSS, inhibiting the expression of proinflammatory factors, improving the expression of anti-inflammatory factors and tight junction proteins, inhibiting the activation of the MAPK and NF-κB signaling pathways, promoting the growth of beneficial microorganisms in the intestine, and inhibiting the growth of harmful microorganisms in the intestine. This study provides a potential therapeutic strategy for the future treatment of IBD patients.

## Figures and Tables

**Figure 1 microorganisms-12-01379-f001:**
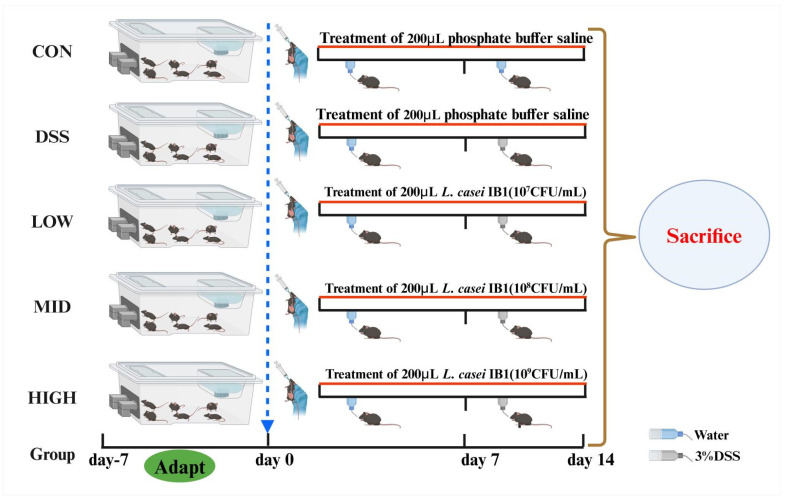
Experimental process and technical route. The mice were given 3% DSS in their drinking water for 7 days before being sacrificed. *L. casei* IB1 was administered daily by oral gavage (200 μL). Graphics created with BioRender. (www.biorender.com), accessed on 18 February 2024.

**Figure 2 microorganisms-12-01379-f002:**
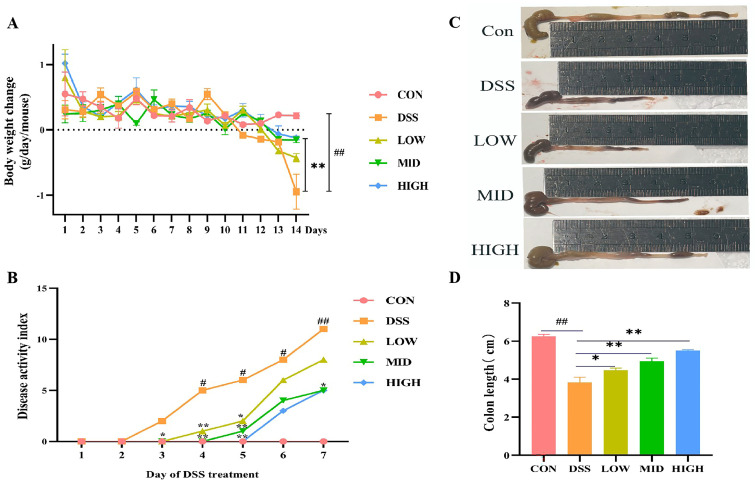
*L. casei* IB1 alleviates DSS-induced colitis in mice. (**A**) Body weight change. (**B**) The DAI score was measured. (**C**) Representative images of colons from mice. (**D**) Statistical analysis of colon length. The data are shown as the means ± SEMs. ^#^
*p* < 0.05, ^##^
*p* < 0.01, compared to the CON group. * *p* < 0.05, ** *p* < 0.01, compared to the DSS group.

**Figure 3 microorganisms-12-01379-f003:**
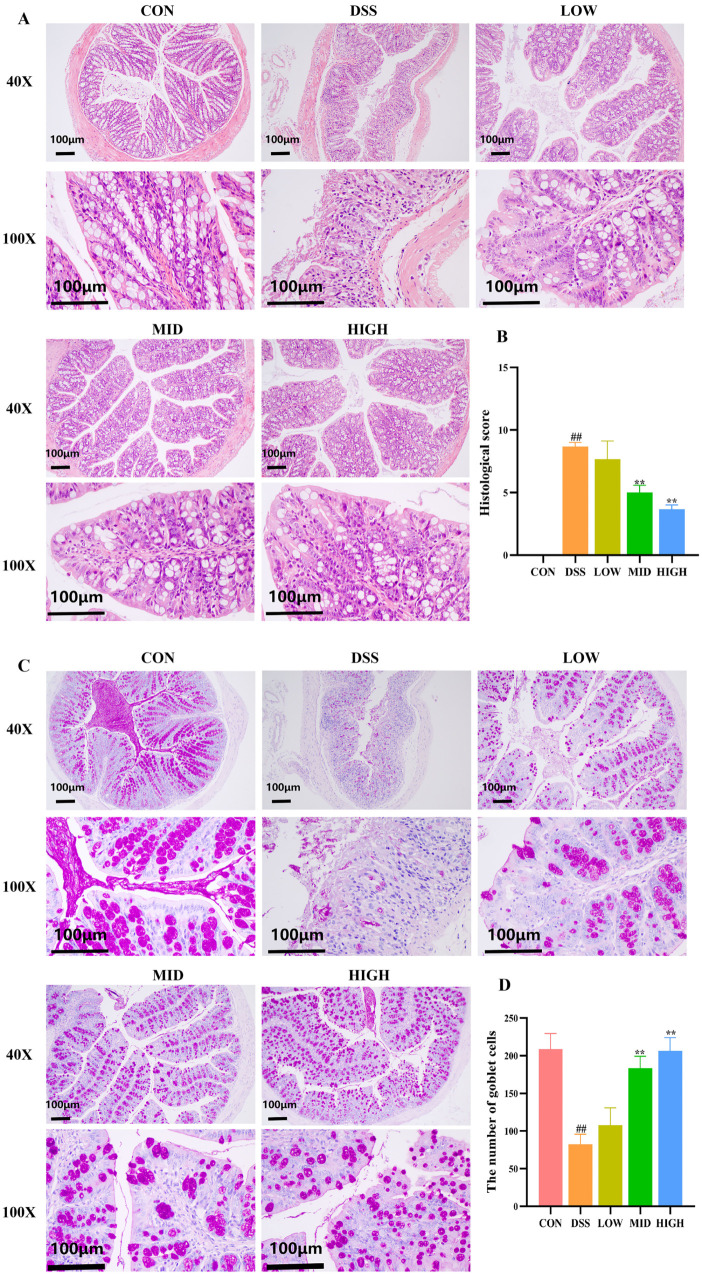
*L. casei* IB1 alleviates DSS-induced colitis in mice. (**A**,**B**) Histopathology scores of the colitis mice post-treatment with *L. casei* IB1. (**C**,**D**) Periodic acid–Schiff staining and the number of goblet cells. The data are shown as the means ± SEMs. ^##^
*p* < 0.01, compared to the CON group. ** *p* < 0.01, compared to the DSS group.

**Figure 4 microorganisms-12-01379-f004:**
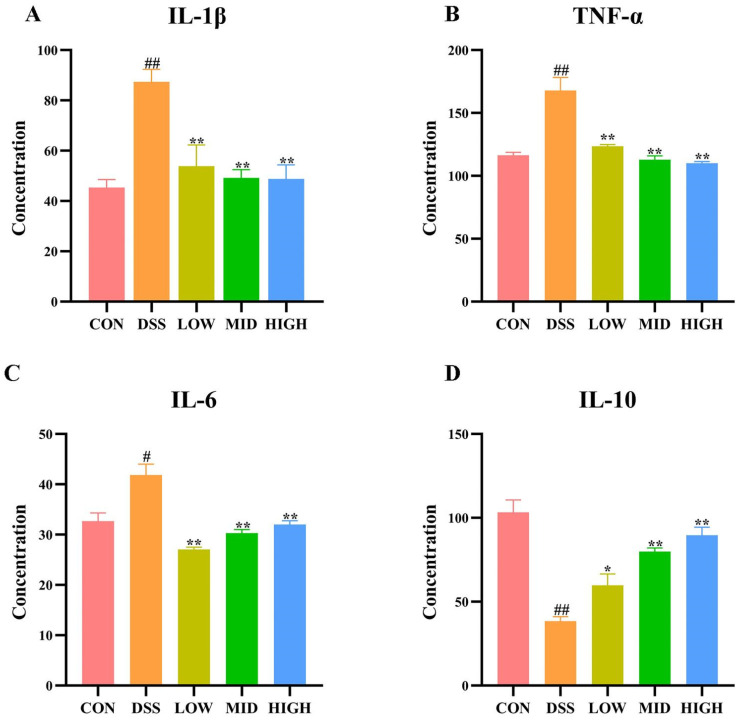
Effect of *L. casei* IB1 on serum cytokine levels. (**A**–**D**) Serum cytokine levels after *L. casei* IB1 treatment. The data are shown as the means ± SEMs. ^#^
*p* < 0.05, ^##^
*p* < 0.01, compared to the CON group. * *p* < 0.05, ** *p* < 0.01, compared to the DSS group.

**Figure 5 microorganisms-12-01379-f005:**
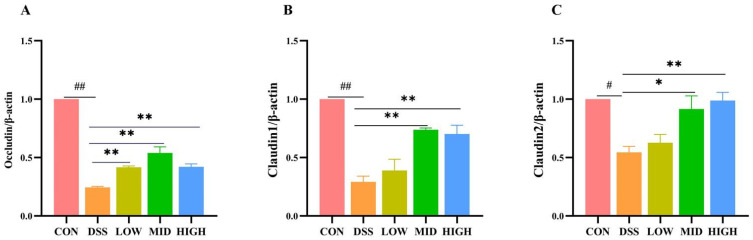
*L. casei* IB1 upregulated tight junction proteins to improve colonic barrier dysfunction in DSS-induced colitis mice. (**A**–**C**) Statistical analysis of Occludin, Claudin1, and Claudin2 levels in the colon. The data are shown as the means ± SEMs. ^#^
*p* < 0.05, ^##^
*p* < 0.01, compared to the CON group. * *p* < 0.05, ** *p* < 0.01, compared to the DSS group.

**Figure 6 microorganisms-12-01379-f006:**
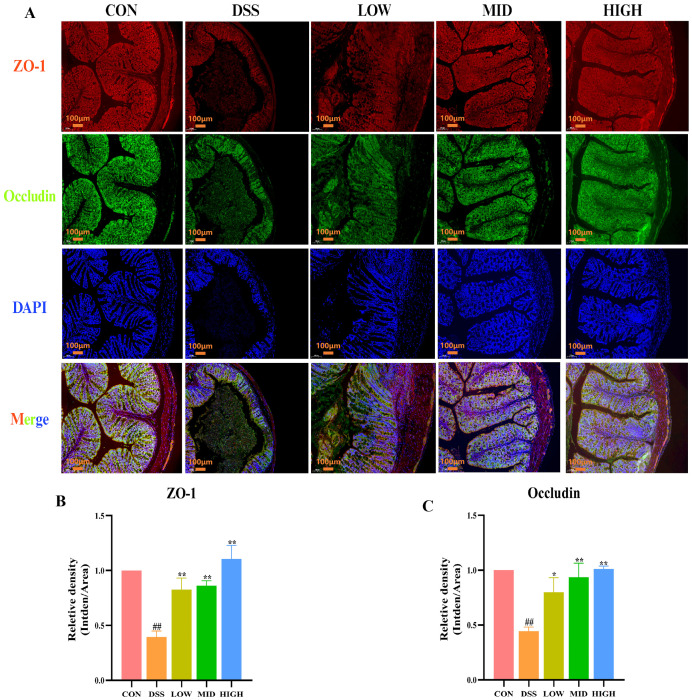
*L. casei* IB1 upregulated tight junction proteins to improve colonic barrier dysfunction in DSS-induced colitis mice. (**A**) Immunofluorescence analysis of ZO-1 and Occludin levels in the colon. (**B**,**C**) Statistical analysis of ZO-1 and Occludin levels in the colon. The data are shown as the means ± SEMs. ^##^
*p* < 0.01, compared to the CON group. * *p* < 0.05, ** *p* < 0.01, compared to the DSS group.

**Figure 7 microorganisms-12-01379-f007:**
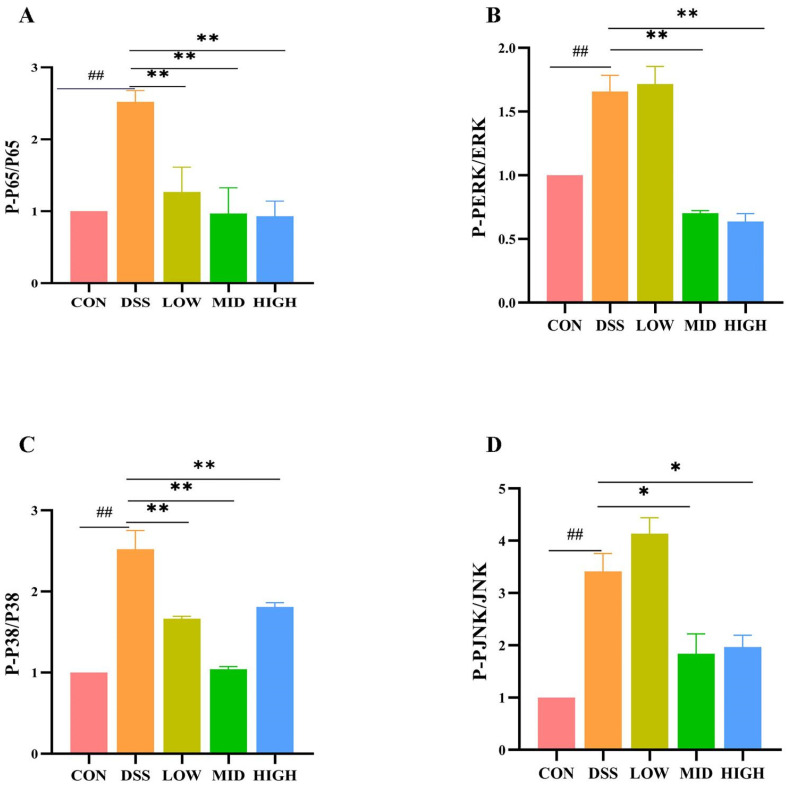
*L. casei* IB1 inhibits the inflammatory response by inhibiting the NF-κB- and MAPK-mediated signaling pathways. (**A**–**D**) Statistical analysis of P65, P-P65, ERK1/2, P-ERK1/2, P38, P-P38, JNK, and P-JNK levels in the colon. The data are shown as the means ± SEMs. ^##^
*p* < 0.01, compared to the CON group. * *p* < 0.05, ** *p* < 0.01, compared to the DSS group.

**Figure 8 microorganisms-12-01379-f008:**
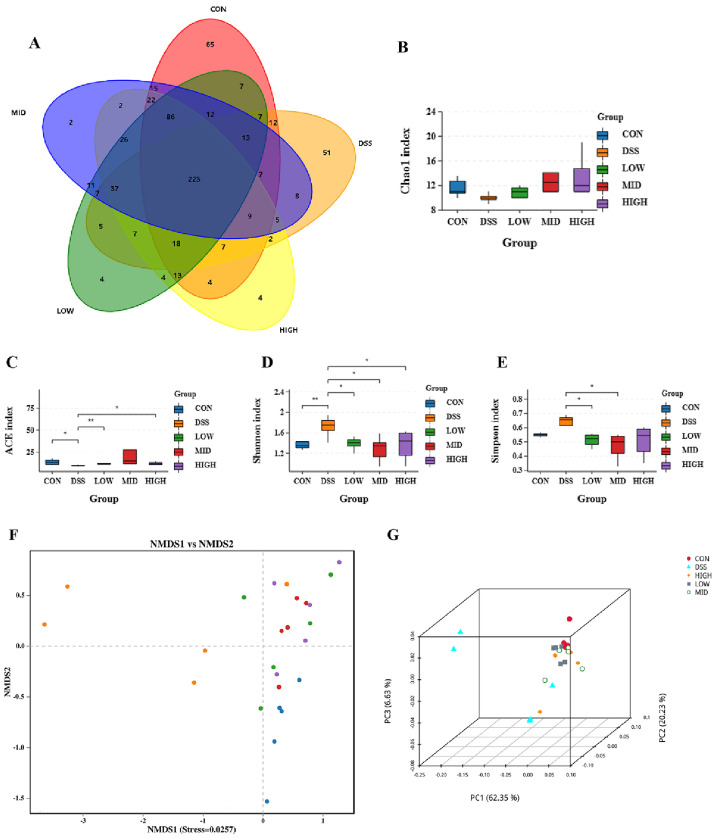
**The** 16S rRNA sequencing results. (**A**) Shared OTU analysis of the content in the colon. (**B**–**E**) α-Diversity analysis using the ACE, Simpson, Shannon, and Chao1 indices. (**F**,**G**) β-Diversity analysis of PCoA and NMDS data. The data are shown as the means ± SEMs. * *p* < 0.05, ** *p* < 0.01, compared to the DSS group.

**Figure 9 microorganisms-12-01379-f009:**
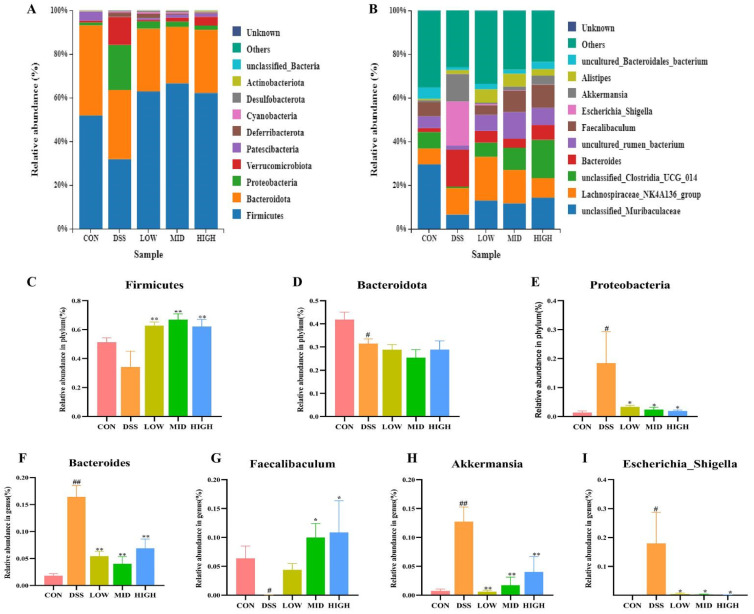
*L. casei* IB1 changed the structural composition of colon microorganisms. (**A**,**B**) Histogram depicting the bacterial species distribution at the phylum and genus levels in the five mouse groups. (**C**–**I**) Statistical analysis of bacterial species at the phylum and genus levels. The data are shown as the means ± SEMs. ^#^
*p* < 0.05, ^##^
*p* < 0.01, compared to the CON group. * *p* < 0.05, ** *p* < 0.01, compared to the DSS group.

**Figure 10 microorganisms-12-01379-f010:**
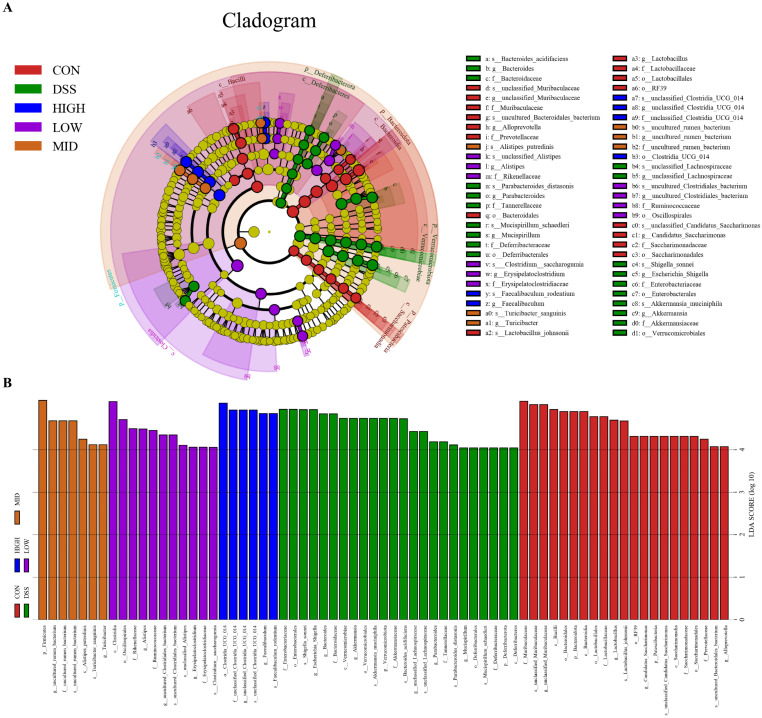
Effects of *L. casei* IB1 on the gut microbiota. (**A**) LefSe multilevel species hierarchy tree. (**B**) The distribution histogram based on LDA.

**Table 1 microorganisms-12-01379-t001:** Disease activity index (DAI) scoring system.

Score	Weight Loss (%)	Fecal Consistency	Blood in Feces
0	None	Normal	Negative (no bleeding)
1	1–5%	Loose stools	Negative
2	5–10%		Hemoccult positive (slight)
3	10–20%	Diarrhea	Hemoccult positive
4	>20%		Gross bleeding

The DAI is a summation of weight loss, fecal consistency, and blood in feces. The DAI was used to evaluate the extent of intestinal inflammation.

**Table 2 microorganisms-12-01379-t002:** Histological colitis scoring system.

Feature Score	Score	Description
Inflammation severity	0	None
	1	Mild
	2	Moderate
	3	Severe
Inflammation extent	0	None
	1	Mucosa
	2	Submucosa
	3	Transmural
Crypt damage	0	None
	1	Basal 1/3 damage
	2	Basal 2/3 damage
	3	Crypt lost, surface epithelium present
	4	Crypt lost, surface epithelium lost
Percent involvement	0	0%
	1	1–25%
	2	26–50%
	3	51–75%
	4	76–100%

Histological criteria used to evaluate the severity of inflammation. The score was made up of the sum of the severity of the inflammation, the extent of the inflammation, crypt damage, and the percentage of involvement.

## Data Availability

The raw data supporting the conclusions of this article will be made available by the authors on request.

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
