# Peer review of "Lacticaseibacillus casei IB1 Alleviates DSS-Induced Inflammatory Bowel Disease by Regulating the Microbiota and Restoring the Intestinal Epithelial Barrier"

_microorganisms, 2024, doi:10.3390/microorganisms12071379_

Round 1

Reviewer 1 Report

Comments and Suggestions for Authors

The work entitled “Lactobacillus casei IB1 alleviates DSS-induced inflammatory 2 bowel disease by regulating the microbiota and restoring the 3 intestinal epithelial barrier”, Outline the probiotic strains of L. casei IB1, especially the anti-inflammatory and barrier protection they provide in a DSS model. Lao et al. have completed a comprehensive work on the well-designed set of experiments. In the DSS model, Autors first carried out the macroscopic, histological, and DAI scores. The study of pro-inflammatory cytokines and pathways in DSS models has been extensive. Experiments on the barrier function, including ZO-1, muc, and occludin genes, are integrated by them. Finally, the microbiome modulation by L. casei IB1 in the DSS-mice model is reported.  What is the novelty of L. casei IB1 compared to other L. casei strains? Any possible molecular effectors found in L. casei IB1, whether a cellular component or a metabolite? I don't have any major concerns to add.

Minor concerns:

Update the taxonomic names of Lactobacillus strains, i.e. Lactobacillus casei should be renamed asLacticaseibacillus casei.

Please provide examples of tight junction proteins that are regulated by microbiota/probiotics in lines 41-42.

Line 84: Please define the abbreviations for DSS+L, DSS+M, and DSS+H when they are cited for the first time.

Figure 1: move to the methods section. IB1 instead of IBL1.

Figure 3. Please add b) and the legend for the axis.

Figure 4: expression levels?

Figure 8: Improve the quality of the figure.

Any hypothesis about increase of Akkermansia in DSS-group?

Reviewer 2 Report

Comments and Suggestions for Authors

REVIEW

Dear authors,

The present work proposes the use of the strain Lactobacillus casei IB1 as a potential treatment of inflammatory bowel disease (IBD) using an animal model of inflammation caused by DSS, the experimental design is adequately planned for the objectives established in the work, while that the methodologies used are described and applied correctly, so the results show supported and justified evidence. However, there are some important deficiencies in the work that must be explained for greater understanding.

Please consider the following comments to improve the content of your manuscript before publication. 

1.       In section 1. Introduction, at no time is the beneficial characteristics of the probiotic strain L. casei IB1 described, since when mentioning that it is a “probiotic” strain, at least background information should be mentioned where it has been evaluated at a level in vitro in some previous work or, failing that, having carried them out in this work. Even if the IB1 strain belongs to a company, it must be indicated which in vitro criteria it passed to be considered probiotic. Add IB1 information to this section.

2.       In section 2.2 Experimental animal design there are several deficiencies, which are:

a)    What is the initial weight of the mice in each group?

b)   How many inocula of strain IB1 (LOW, MID, HIGH) were administered throughout the experiments? In the diagram of Figure 1 it is not understood.

c)    On the 15th day of the experiment (sacrifice) was IB1 still administered or was some time allowed to elapse between the last administration of IB1 and sacrifice?

d)   At least a control group of the IB1 strain (107, 108 or 109 CFU/mL) without DSS should have been considered, for the safety evaluation of L. casei IB1.

3.       In section 4. Discussion I consider that the immunomodular effect is not given by the administration of IB1, but by the modification it exerts on the diversity and abundance of the microbiota during the period of induction of colitis by DSS, and these genera (different from Lactobacillus) are those that modulate the interaction with the host elements (intestinal epithelium and lamina propria) so that they trigger a type of controlled immune response. The translocation of commensal microorganisms with pathobiont potential to the lamina propria can also explain the profile of cytokines present in circulation, so these elements must also be considered in the Discussion.

4.       Figures 3A, 3C, 6A, 8A-G and 9A-C should be enlarged because they cannot be seen well.

5.       Figures 5A and 7A can be placed as supplementary material, just leave the bar graphs.

6.       Histopathology of important organs such as the liver and spleen was missing.

7.       Figure 1: correct the name of the strain “IBL1” in the LOW, MID and HIGH groups.

8.       Figure 3B: write the name “Histopathology score” on the “Y” axis and add the letter “B”.

9.       Figure 4A-D: they did not evaluate the expression of the cytokines IL-1b, TNF-a, IL6 and IL-10, therefore the “Y” axis should say “Concentration”.

10.     There are spelling mistakes that must be corrected and the name of some microorganisms written in italics.

Please amend the requested comments and submit the revision file.

Reviewer 3 Report

Comments and Suggestions for Authors

Dear Sirs,

The authors of this study present interesting data elucidating the potential role and underlying mechanisms through which Lactobacillus casei IB1 mitigates DSS-induced IBD. While the results of their study are highly intriguing, in my opinion, there is room for some improvements.

- Introduction

1.Lines 28-29: I would rephrase as follows: Inflammatory bowel disease (IBD), comprising ulcerative colitis (UC) and Crohn's disease (CD)[1-3], is an idiopathic, chronic, and frequently recurrent condition impacting the gastrointestinal tract, with global implications for human health.

2.The authors could integrate a paragraph discussing the role of probiotics in IBD, drawing from recent manuscripts such as those cited (PMID: 38585636, PMID: 35628274). Additionally, they could provide evidence regarding the specific involvement of Lactobacillus in inflammatory bowel disease, as indicated by the literature (PMID: 37773196). This approach would effectively introduce readers to the subject matter of their study and contextualize its significance within the broader field of IBD research.

- Discussion:

1.Lines 340-345: Please rephrase as follows:

Therefore, we used a DSS-induced mouse colitis model to study the effect of L. casei IB1 on colitis in C57BL/6 mice. After 7 days of 3% DSS treatment, the mice showed severe damage, and the clinical symptoms were weight loss, diarrhea, and rectal bleeding [21, 42, 43]. In addition, we confirmed that DSS significantly increased the DAI. On the contrary, the symptoms of colitis in mice treated with 343 L. casei IB1 were significantly alleviated. Compared with the DSS group, treatment with L. casei IB1 significantly increased the colon length of the mice.

2.Lines 347-348: please rephrase: crypt injury

3.Lines 355-357: Maybe the authors mean: They modulate cell differentiation and growth by binding to corresponding receptors and regulating immune responses. ?

4.Lines 363-365: Please rephrase: In this study, we measured the serum levels of pro-inflammatory factors (TNF-α, IL-1β, IL-6) and the anti-inflammatory factor IL-10 using ELISA. Compared to the DSS group, the levels of pro-inflammatory factors were significantly decreased, while the level of IL-10 was markedly increased after treatment with L. casei IB1.

5.Line 376: Please use ‘’mitigated’’ instead of alleviated, as you have used the same word before.

6.The authors should incorporate a paragraph at the end of the discussion section discussing the possible limitations and the strengths of their study. Summarizing in this way will clarify why their study merits publication.

Comments on the Quality of English Language

Moderate editing of English language suggested. 

Round 2

Reviewer 2 Report

Comments and Suggestions for Authors

REVIEW

Dear authors,

The suggested corrections and modifications were made, which better supports the research.

Please consider the following comments to improve the content of your manuscript before publication. 

Line 57: write the abbreviation “IB1” in capital letters and the term “in vivo” in italics.

Line 96: separate with a space “mLof”.

Line 355: write in cursive “L. casei”.

Line 431: write the name of the species in cursive letters “rodentium”.

Line 440: remove a space in “Shigella  decreased”.

Line 441: write the term in cursive letters “in vivo”.

Line 442: write the term in cursive letters “in vitro”.

Line 447: separate with a space “[71].Therefore”.

Please amend the requested comments and submit the revision file.
